# Heuristics for the sustainable harvest of wildlife in stochastic social-ecological systems

Elizabeth A. Law[1]*, John D. C. Linnell[1,2], Bram van Moorter[1], Erlend B. Nilsen[1,3]

**1** Norwegian Institute of Nature Research (NINA), Trondheim, Norway, **2** Department of Forestry and Wildlife Management, Inland Norway University of Applied Sciences, Evenstad, Norway, **3** Faculty of Biosciences and Aquaculture, Nord University, Steinkjer, Norway

* workingconservation@gmail.com

**Data Availability Statement:** Input data, simulation code and results are available in OSF repository https://osf.io/u52rp/?view_only=e36abdca3e3c45d8813e6f7b20ce159a Analysis code and results are available in OSF repository

## Abstract

Sustainable wildlife harvest is challenging due to the complexity of uncertain social-ecological systems, and diverse stakeholder perspectives of sustainability. In these systems, semi-complex stochastic simulation models can provide heuristics that bridge the gap between highly simplified theoretical models and highly context-specific case-studies. Such heuristics allow for more nuanced recommendations in low-knowledge contexts, and an improved understanding of model sensitivity and transferability to novel contexts. We develop semi-complex Management Strategy Evaluation (MSE) models capturing dynamics and variability in ecological processes, monitoring, decision-making, and harvest implementation, under a diverse range of contexts. Results reveal the fundamental challenges of achieving sustainability in wildlife harvest. Environmental contexts were important in determining optimal harvest parameters, but overall, evaluation contexts more strongly influenced perceived outcomes, optimal harvest parameters and optimal harvest strategies. Importantly, simple composite metrics popular in the theoretical literature (e.g. focusing on maximizing yield and population persistence only) often diverged from more holistic composite metrics that include a wider range of population and harvest objectives, and better reflect the trade-offs in real world applied contexts. While adaptive harvest strategies were most frequently preferred, particularly for more complex environmental contexts (e.g. high uncertainty or variability), our simulations map out cases where these heuristics may not hold. Despite not always being the optimal solution, overall adaptive harvest strategies resulted in the least value forgone, and are likely to give the best outcomes under future climatic variability and uncertainty. This demonstrates the potential value of heuristics for guiding applied management.

## Introduction

Harvest is one of the most common forms of management for many wildlife species [1, 2]. Wildlife harvest is important socially, culturally and economically, both for creating direct benefits (e.g. meat, income, recreation, tradition) and to avoid costs due to human-wildlife conflicts (e.g. vehicle collisions, predation on domestic animals, and competition or pathogen

https://osf.io/cgwa6/?view_only=973dda4c88ea4a008c3b6e58ff149822.

**Funding:** This study was funded by the Research Council of Norway (https://www.forskningsradet.no/; grant 251112; JL, BM, EN). The funders had no role in study design, data collection and analysis, decision to publish, or preparation of the manuscript.

**Competing interests:** The authors have declared that no competing interests exist.

spread between wild and domestic stock) [1, 3–5]. Because of their socio-economic and ecological importance, wildlife-harvest systems are typically managed with an overarching aim of sustainability [6]. Yet 'sustainability' is a multi-faceted, ill-defined, and evolving term: whilst the early optimal harvest literature focused on ensuring persistence of the species and maximal harvests, contemporary perspectives on sustainability encompass diverse economic and social concepts, ecological, habitat, and ecosystem-based criteria, and precaution under uncertainty [7, 8]. This includes an increasing appreciation of diverse stakeholder perspectives (i.e. social equity) [9, 10], animal welfare, animal rights, and 'compassionate' conservation [11, 12].

Under the lens of these complexities and stakeholder conflicts, it is no surprise that concepts of sustainability are often poorly applied in wildlife harvest systems [6]. Established theory on optimal harvest strategies can often seem highly abstract through a focus on limited objectives, typically maximization of harvest volumes without sacrificing population persistence [13–16]. More recently, theoretical objectives have included variability of population sizes and harvest [17]. While some highly detailed applied models exist [e.g. 2, 5, 18–21], in many cases these are unavailable: many wildlife management systems lack all but the most rudimental parameters, due to limited resources and poorly developed institutional frameworks [6, 22]. In practice determining quotas in terrestrial systems is often an inexact, adaptive science at best [23]. Further, even in the best studied cases, important elements of the social-ecological system remain uncertain or contested [24–28], and there is a trend away from defining single 'best' harvest strategies, in order to avoid over-reliance on models with assumptions that may not hold in reality [29].

Heuristics are practical and accessible guidelines designed to give good 'rules-of-thumb', e.g. management recommendations that lead to good outcomes over a wide range of cases and contexts [30]. In a wildlife management context, heuristics developed from semi-complex case studies can bridge the gap between highly simplified models developed to demonstrate theory, and highly context-specific case studies [31]. Benefits to addressing this space are three-fold. First, more nuanced heuristics can be developed for application in knowledge-poor contexts [30]. This is required in wildlife harvest because in most cases the socio-ecological contexts are more complex than those addressed by existing theoretical models. From an implementation perspective, managers are also more likely to accept and utilize evidence that is more specific to their context [32–34]. Second, heuristics can help to guide sensitivity analyses in knowledge-rich contexts, where complex case-study models can be developed, but the range of parameters is too great for a meaningful development or interpretation of a global sensitivity analysis [35]. Third, heuristics can improve the understanding of context comparability. Causal inference, i.e. where specific causal impacts can be robustly identified (e.g. through analysis of pairwise comparisons in which only the variable of interest changes) is challenged in complex socio-ecological contexts such as wildlife harvest due to the low number of comparable empirical examples to study [36], and this often results in comparisons across contexts [33]. Heuristics at semi-complex levels can give us knowledge on the potential comparability of different contexts, and thereby inform the appropriate transfer of causal inference estimates across different contexts [37].

Heuristics can be derived by induction from empirical experience, or by deduction from simulation models [38, 39]. However, it is challenging to robustly derive heuristics from empirical case studies in wildlife harvest, even with meta-analyses, because of the conceptual, logistical, and ethical difficulty in conducting systematic experimentation at the scales required (i.e. while it is theoretically possible, there are practical challenges in collating enough comparable studies for comparisons without confounding factors complicating interpretation) [36, 40]. As a result, mathematical and stochastic simulation models are well established in the conservation and wildlife-management literature. Typical simulation models focus on stochastic

population dynamics, for example applied in population viability analysis [6, 21, 41]. In traditional harvest models, population dynamics is coupled with harvest to assess how variation in harvest intensity affects population persistence and harvest off-take [14, 16, 17]. Management Strategy Evaluation (MSE) models expand from these, encompassing stochastic simulations of management in socio-ecological systems incorporating a more holistic set of ecological and social components [42]. MSE models are well established in fisheries [43] and increasingly used in terrestrial management scenarios, typically as highly detailed case study simulations [e.g. 2, 5, 18–21]. MSE models have been used to address key knowledge gaps regarding the implications of uncertainty in the multiple socio-economic facets of wildlife harvest systems [3], and allow levels of systematic assessment impossible in real-world experiments. From fisheries management systems, literature syntheses of MSE case studies that contrast different harvest strategies suggest strong context-dependencies of optimal strategies [39]. No such synthesis has been conducted for terrestrial systems.

To develop heuristics for sustainable terrestrial wildlife harvest, we constructed a semi-complex MSE framework that allowed us to assess sustainability under a range of environmental contexts and from diverse socio-ecological perspectives, and thereby identify patterns of context dependencies over environmental and evaluation contexts. We simulate a set of species (moose, *Alces alces*, roe deer, *Capreolus capreolus*, and willow ptarmigan, *Lagopus lagopus*) from across the fast-slow life-history gradient, a commonly used heuristic for theory development in wildlife demography describing patterns of covariation in life-history traits across body size, longevity, and fecundity [44, 45]. In contrast to most previous harvest system models that focus on a narrow set of objectives, we evaluate sustainability over 10 evaluation metrics reflecting volumes and variability of both population abundance and harvest, combined into 6 stakeholder perspectives relevant for terrestrial contexts. In this way, our working definition of sustainability operates at the system level and aims to capture both outcomes and potential conflicts between stakeholders that may moderate these. To simulate the variability often inherent in socio-ecological systems, we include multiple types of variability [46] representing both temporal stochasticity, as well as parameter uncertainty related to monitoring, management decision, and harvest implementation components. This MSE framework bridges a gap between simplified harvest models with a narrow focus on harvest off-take and highly context-specific applied case studies, with the intention of producing heuristics that are directly applicable to real-world settings. We compare the 289,848 simulation models to uncover: 1) How do wildlife harvest outcomes differ in different contexts? 2) How do different contexts influence optimal harvest parameters in the different systems? 3) Which harvest systems are optimal in different contexts? 4) How much can decision-making improve through integration of environmental and evaluation context-specific heuristics?

## Materials and methods

We develop a MSE model that generalises a terrestrial wildlife-harvest system, with components of 1) resource dynamics, 2) monitoring observations, 3) quota setting, 4) harvest implementation, and 5) sustainability evaluation. Simulations occur in yearly time steps ($t$), across a time series of 20 years (broadly considered long term for applied management plans), with multiple replications ($i = 1000$) per scenario. Full model description and parameter values are available in S1 Appendix, and summarised here.

### MSE framework

The MSE framework developed here consists of five main components (Fig 1), representing the main components of a socio-ecological harvest system. The **resource component** simulates

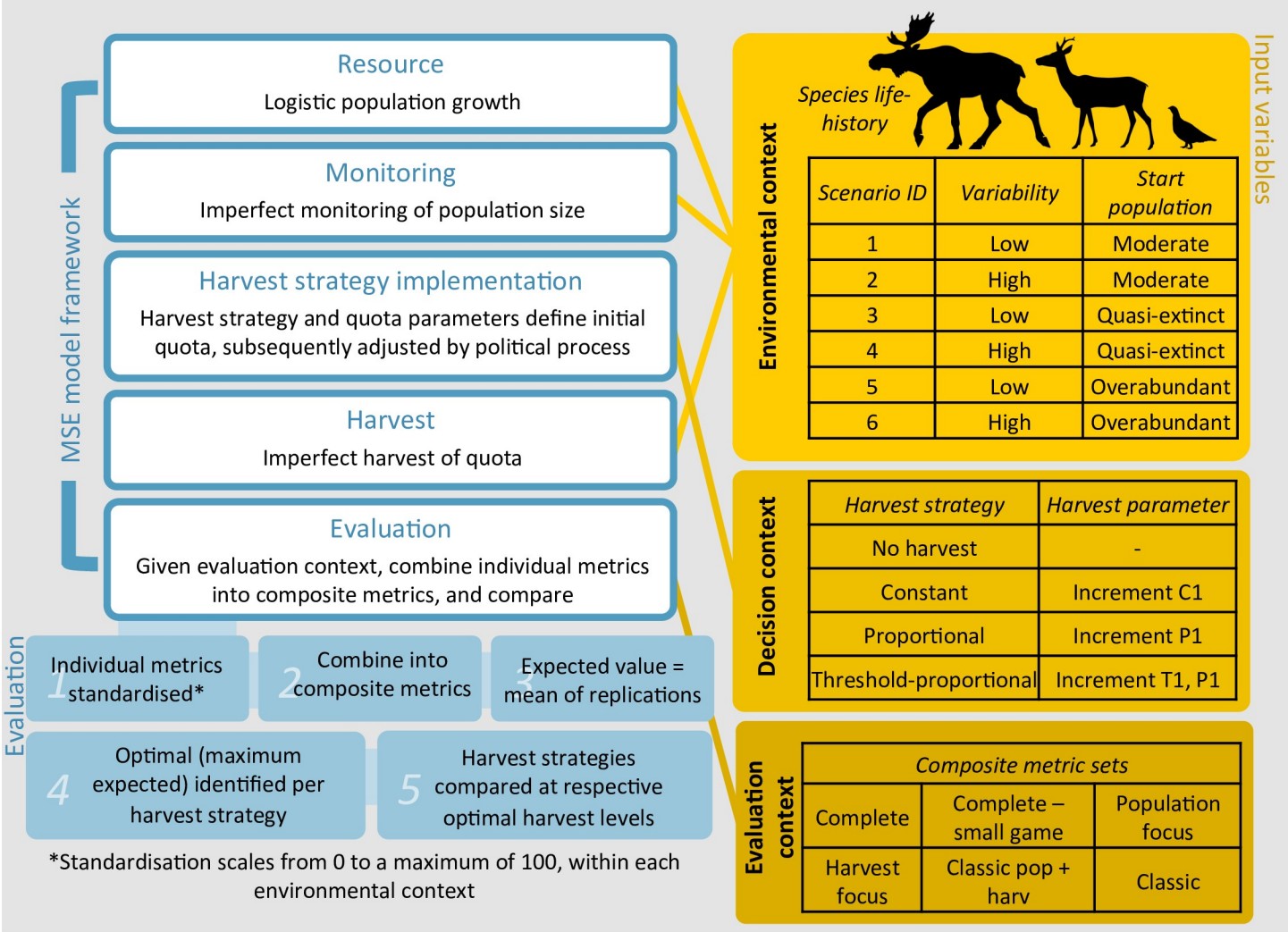

**Fig 1. MSE framework.** The Management Strategy Evaluation (MSE) model simulates a wildlife harvest system over a 20 year timeframe, with each environmental and decision context including 1000 stochastic replications. Evaluation contexts are simulated through combinations of different evaluation metric sets. Species types span a fast-slow life-history gradient, determining growth rates and carrying capacity, variation levels in growth rates and monitoring variability, and critical thresholds. Stochastic parameters simulate yearly stochasticity and iteration level uncertainty. A full description of the model and parameter values are specified in S1 Appendix.

growth of a population $N_{i,t}$, using logistic growth determined by the population's intrinsic growth rate, $r_{i,t}$, and carrying capacity, $K$. The **monitoring component** is simulated by a single variation factor ($m_{i,t}$) acting on $N_{i,t}$, to give an estimate of the population size ($\hat{N}_{i,t}$), to be used as the basis for management decisions. The **management-decisions component** comprises two parts. First, a harvest strategy is applied, converting $\hat{N}_{i,t}$ into an initial quota, $Q_{i,t}$, given a set of quota parameters. $Q_{i,t}$ is then subject to random variation ($q_{i,t}$) to simulate variability of stakeholder influence during the quota setting process, to give a modified quota $Q'_{i,t}$. The **harvest implementation component** simulates imperfect harvest implementation, effected as variation ($h_{i,t}$) around $Q'_{i,t}$ to give the realised harvest ($H_{i,t}$). This amount is then removed from $N_{i,t}$, before continuing to the next timestep. Stochastic parameters include $r$, $m$, $q$, and $h$, which simulate environmental stochasticity, imperfect implementation, and parameter uncertainty. We assumed that the uncertainty followed a normal distribution, partitioned over years ($t$)

and replications (*i*).The **evaluation component** occurs after each simulation is complete, calculating performance metrics of each iteration over the entire timeframe, and summarising over replications in the scenario run (see details below and in S1 Appendix).

### Environmental context and decision variable parameters

In our modelling framework, species life-history, level of environmental variability and parameter uncertainty, and starting population scenarios collectively represent the **environmental context** within which the simulation takes place. We simulate three species spanning a slow-fast life-history gradient of common game species (S1.1 Table in S1 Appendix). The species are based on wildlife harvested in a Norwegian context, but with global relevance. The moose (*Alces alces*) is a large ungulate, with a relatively low growth rate, carrying capacity, monitoring variation, and critical thresholds for evaluating population size. The roe deer (*Capreolus capreolus*) is a small ungulate with a moderate growth rate, carrying capacity, monitoring variation, and critical thresholds. The willow ptarmigan (*Lagopus lagopus*) is a game bird with a large potential growth rate, carrying capacity, monitoring variation, and critical thresholds.

For each species, variability and starting population scenario combinations are identified numerically (SID 1–6) defined in Fig 1. We simulated two variability scenarios, where variability in *r*, *m*, *q*, and *h* was either all *low* (SID 1,3,5) or all *high* (SID 2,4,6), to represent systems with different extremes of variability and parameter uncertainty. Each of these species -variability scenarios were coupled with three distinct scenarios for the population size at the start of the simulation period: 1) *moderate* (best case), i.e. the midpoint of low and high critical thresholds (SID 1,2), 2) *quasi-extinction* (SID 3,4), and 3) *overabundance* (SID 5,6). Alternative starting populations test the robustness of the harvest strategies to extreme perturbations in population size, as well as being relevant for special management cases (e.g. overabundant species, or recovery of endangered species into harvestable populations). In total, we evaluated 3 species × 2 variability × 3 starting population size scenarios, yielding a total of 18 environmental contexts.

For each of these environmental scenarios, we evaluated a range of harvest alternatives. The 4 *harvest strategies* and their respective range of *harvest parameters* together represent **decision variables**. We define the harvest strategy to include *constant* harvest (a set number of individuals harvested yearly), *proportional* harvest (a set proportion of the population harvested yearly), *threshold-proportional* harvest (a set proportion of the population taken yearly, provided the population is above a certain threshold), and a *no harvest* baseline. Harvest parameters define the intensity of harvesting under a given harvest strategy. For example, for constant harvest, the 'constant' parameter specifies the fixed annual quota size, and for proportional harvest the 'proportion' parameter specifies the harvest fractions. We searched across a wide range of constants, proportions, and thresholds in order to identify and compare optimal strategies across a diversity of potential objectives. Constants were varied in increments of 1% of the respective species 'moderate' population sizes, proportions were varied in 1% increments, and thresholds also in 1% increments ranging from the respective species 'quasi-extinction' population threshold, to a 'moderate' population size (see S1.2 Table in S1 Appendix). Within one simulation, the harvest strategies and parameters remain consistent throughout the timeframe, although the simulated harvests themselves vary due to variability in quota setting, available population size, and harvest imperfections. Thus, the 18 environmental contexts were each subject to 4 harvest strategies, totalling 72 environment × harvest strategy cases. Overall, we simulated 289, 848 different environment × harvest strategy × harvest parameter models, each with 1000 iterations.

## Evaluation contexts

In our MSE framework, **evaluation contexts** are designed to reflect different stakeholder values and perspectives relevant to terrestrial wildlife harvest scenarios. We first define 10 *individual metrics* representing different stakeholder objectives over various socio-ecological and harvest-based sustainability objectives (Table 1), and then combine them into six *composite scores* representing alternative evaluation contexts with different emphases (Table 2). While we aimed to include as broad a range of sustainability metrics as possible, these may not be adequately representative of ecological or socio-economic objectives that are not a function of volumes or variability of either harvest or population sizes, for example spatial, sex, and age distribution of individuals and consequent effects on ecological functions and equity of opportunity. We standardise each individual metric so that 0 represents the worst score (e.g. zero years of stable population, a mean harvest of zero, or the maximum observed harvest variability), and 100 represents the most desirable expected outcome possible (e.g. all years with stable population, zero harvest variability, or the largest observed harvest) over all replications and decision variables for each respective environmental context. Full details and summaries of raw and transformed scores are provided in S1 Appendix.

Evaluation contexts are represented by the composite scores via the individual metrics contributing to the composite score. These range from a *complete* set including all metrics, to a *classic* set that includes metrics most commonly included in the classic theoretical literature, namely maximize harvest and population persistence. Other sets represent particular contexts, such as a focus only on *population* or *harvest* related metrics. Composite metrics are the mean

**Table 1. Individual sustainability metrics.**

| Objective group | Objective | Criteria | Code |
|---|---|---|---|
| *Persistence* | **Avoiding extinctions.** A fundamental objective of ecological and economic sustainability. | For individual replications, this is a binary score (0 = extinction, 1 = persistence of the population over the time frame). This is averaged over replications as a probability. | *persistence* |
| *Population* | **Population stability.** Avoiding population extremes. | Number of years population remains between *high* and *low* critical thresholds | *stable population* |
| | **Avoiding low or functionally extinct populations.** To provide adequate populations for harvest, ecological functionality, and buffer against extinctions. | Number of years population remains above the *quasi-extinction* critical threshold | *above quasi-extinct* |
| | | Number of years population remains above the *low* critical threshold | *above low* |
| | **Avoiding high and overabundant populations**. To minimize wildlife conflict and ecological damage from overabundant populations. Note, this may not be a concern for small game species. | Number of years population remains below *high* critical threshold | *below high* |
| | | Number of years population remains below the *overabundance* critical threshold | *below overabundant* |
| *Harvest* | **Mean annual harvest.** To provide the maximum opportunity for economic and social benefits of harvest. | Mean yearly harvest | *harvest mean* |
| | **Minimum harvest** experienced across the timeframe. To maximize harvest opportunity over every point in the timeframe. | Minimum harvest size across the timeframe | *harvest minimum* |
| | **Avoiding years experiencing zero harvest.** To provide consistency of harvest experience and income for harvesters and associated economies. | Number of years harvest is not zero | *harvest non-zeros* |
| | **Limiting harvest variability.** While some variability may be accepted as an inevitability in variable contexts, consistency of harvest improves predictability and the consistency of capital required for its implementation. | 0 –Standard deviation of harvests over the timeframe | *harvest consistency* |

Sustainability metrics used in this analysis represent a wide variety of common stakeholder concerns, and include fundamental sustainability objective of persistence, as well as other *population-based* and *harvest-based* metrics. Here they are defined so that, within each metric, higher scores are more desirable.

**Table 2. Composite metrics.**

| Composite metric set | Individual metric | | | | | | | | | |
|---|---|---|---|---|---|---|---|---|---|---|
| | Persistence | Above quasi-extinct | Above low | Stable population | Below high | Below overabundant | Harvest mean | Harvest minimum | Harvest non-zeros | Harvest consistency |
| Classic harv. | ✓ | ✗ | ✗ | ✗ | ✗ | ✗ | ✓ | ✗ | ✗ | ✗ |
| Classic pop. +harv. | ✓ | ✗ | ✗ | ✓ | ✗ | ✗ | ✓ | ✗ | ✗ | ✗ |
| Population focus | ✓ | ✓ | ✓ | ✓ | ✓ | ✓ | ✗ | ✗ | ✗ | ✗ |
| Harvest focus | ✓ | ✗ | ✗ | ✗ | ✗ | ✗ | ✓ | ✓ | ✓ | ✓ |
| Complete (small game) | ✓ | ✓ | ✓ | ✗ | ✗ | ✗ | ✓ | ✓ | ✓ | ✓ |
| Complete | ✓ | ✓ | ✓ | ✓ | ✓ | ✓ | ✓ | ✓ | ✓ | ✓ |

Composite metrics are comprised of six different *sets* of individual metrics designed to reflect alternative evaluation perspectives. Inclusion in sets is denoted by a tick (included) or cross (not included); included metrics are averaged to give the composite score.

score of the set of individual metrics from which it is comprised. Due to co-dependencies among individual metrics, composite scores are first calculated for each replicate, before averaging over each harvest scenario. As a side note, this is equivalent to a risk neutral expectation of a utilitarian 'aggregate benefit' ethic, and ensures composite scores remain on a similar scale when involving different numbers of individual metrics. Composite metric scores therefore represent outcomes as perceived under specific stakeholder contexts, but simplistically assume that these individual metrics represent stakeholder utility, that individual metric utilities are equivalent and substitutable, and that aggregate utility is reflected through the average of the individual metrics, and that stakeholders display linear preferences.

## Comparative analysis, heuristics, and potential improvement in decision-making

We sought heuristics for a) determining the likely impacts of environmental and evaluation contextual factors, and b) choosing optimal harvest parameters or strategies, based on the expected (i.e. average) composite metric scores. This assumes a 'benevolent decision-maker' basing their decisions on a rational, risk-neutral optimization of the composite score. Assuming the composite score could be an accurate reflection of social utility, this reflects the potential for stakeholders to be satisfied with the respective outcome. Use of a semi-complex MSE model with the same framework across multiple environmental and evaluation contexts allows a full factorial design in which pairwise comparisons can be made between models that are the same in every way except for the variable of interest. Overall, we compared 18 environmental contexts × 4 harvest strategies × a custom range of harvest parameters × 6 evaluation contexts, totalling 108 environmental × evaluation contexts, 432 environmental × harvest strategy × evaluation contexts, selecting optimal harvest parameters and strategies or each of the evaluation contexts from the 289,848 environmental × decision variable (i.e. harvest strategy and parameters) contexts.

If more information is known (e.g. the environmental context, or the evaluation context), decision-makers are likely to be able to make more appropriate decisions within that context. This is not always the case, however, for example if the same strategy is chosen regardless of the availability of the information. We quantify potential improvement in decision making effected through the use of context-specific heuristics, versus a generalised heuristic, using both the relative frequency of the chosen strategies being optimal, as well as the average value forgone. Value forgone represents the difference in composite score value achieved when

using a (potentially suboptimal) strategy within a specific context, compared to the optimal strategy for that respective environmental and evaluation context. If a strategy is suboptimal, potential value forgone can range from negligible, to 100% of the optimised value.

### Code and data availability

We constructed the model in R [47], using tidyverse [48] and truncnorm [49], parallelized with doSNOW [50]. For graphics, we used ggplot2 [51], ggtable [52], cowplot [53], and magick [54]. For links to all data, code, and results, see data availability statement.

## Results

### Composite scores

Composite scores show considerable overlaps in outcomes between the various harvest strategies and parameters (Fig 2). In general, suboptimal harvest strategies with optimised harvest parameters can often perform better than optimal harvest strategies with poorly selected harvest parameters (Fig 2). This was even more clear when considering potential variability (S2.1 Section in S2 Appendix). Overall, only 11% of the 432 environmental and evaluation context combinations had composite metric scores of over 85% (i.e. performing well across all included metrics), highlighting that poor performances are common, and conflicts between individual metrics, and thus between stakeholder interests, are likely in terrestrial harvest management. Contexts resulting in high composite scores were typically related to relatively stable environments, adaptive harvest strategies (i.e. proportional or threshold proportional), and for evaluation contexts with a population metric focus. Only 4 cases achieved expected maximum scores of 100% (Fig 3); these included *threshold proportional* harvest for *moose* in SID1 and SID 2, and *roe deer* in SID 1, and *proportional* harvest for *moose* in SID 1, all based on the *population focus* evaluation context.

To determine the impact of environmental (i.e. species, variability and starting population size) and evaluation context factors (composite metric types), we assessed the pairwise contrasts between simulations varying only in terms of each specific factor respectively (Fig 4). For the majority of the pairwise contrasts, faster life history species, extreme starting population sizes, and higher variability scenarios result in lower composite metric scores, indicating either irresolvable poor scores and/or stronger conflicts between objectives. However, there are exceptions to these general patterns for most contrasts (Fig 4). Contrasts between evaluation contexts are less predictable, as these scores reflect the number of metrics included, as well as their themes. More complex composite metrics that include more individual metrics were often higher scoring than simpler metrics. For example, the *Complete* set typically scored higher than *Classic pop.+harv.* (true for 94% of the pairwise contrasts). This occurs because the majority of the additional metrics in more complete sets were often less conflicting than those included in the classical sets. Overall, the *Population focus* set was the highest scoring in the majority of pairwise contrasts, likely reflecting the lack of conflict with harvest objectives.

### Optimum harvest parameters

Optimum harvest parameters (that maximize the composite metric score) varied across environmental and evaluation contexts (S2 Appendix). Within a given harvest strategy, different environmental and evaluation contexts had most influence on optimal parameter values (Fig 5). For instance, starting population size was the most universally important determinant for the score within constant harvest strategies (Fig 5A). Higher variability typically decreased the optimal constant harvest rate, whereas optimal constant harvest rates did not vary much

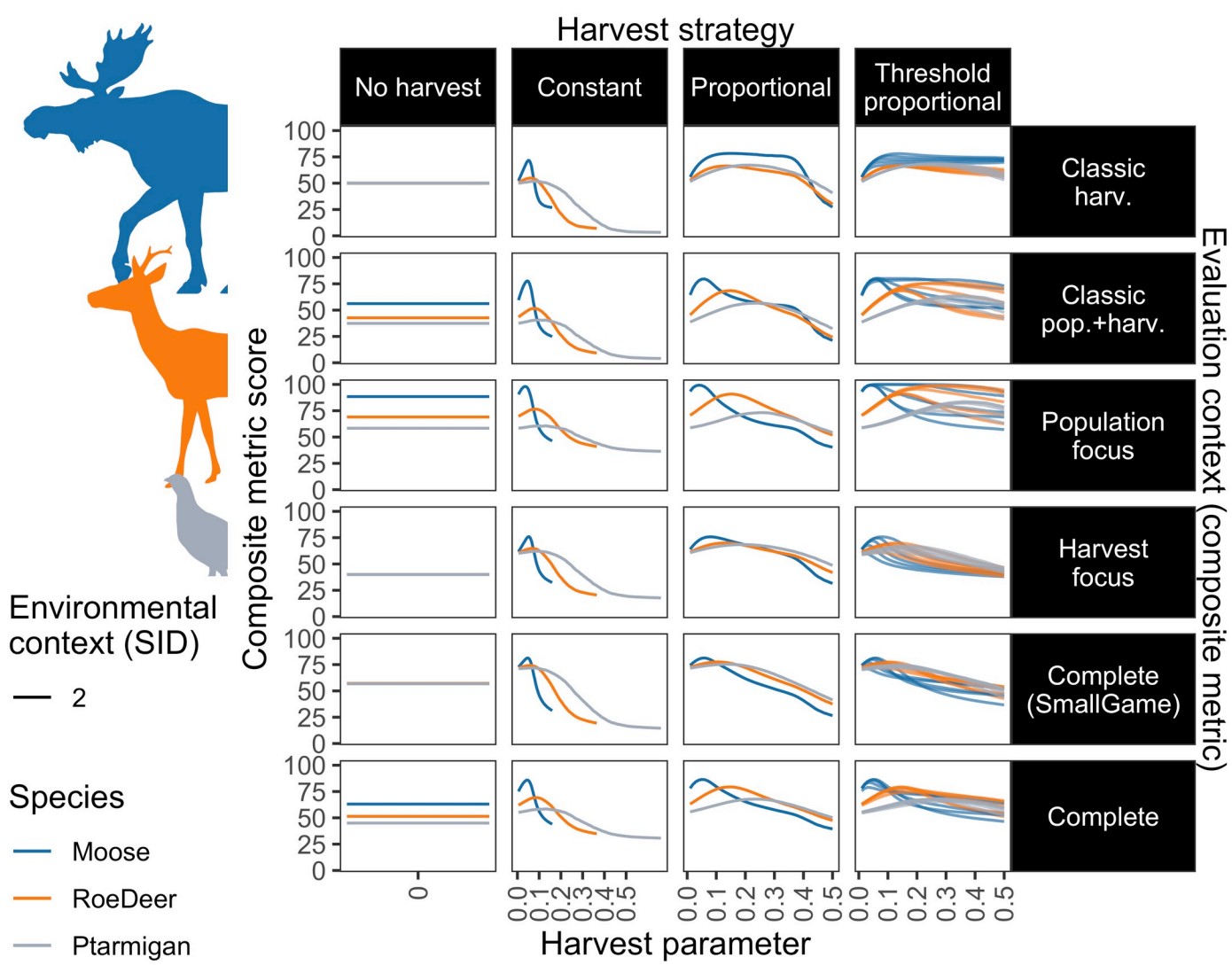

**Fig 2. Composite scores across harvest strategies and parameters.** Composite scores (y-axis) for each composite metric set (panel rows), under each harvest strategy (panel column) and harvest parameter (x-axis). For the constant harvest strategy (second column), the x-axis shows the constant scaled by the maximum constant per species. For the threshold proportional strategy (fourth column), the x-axis shows the proportion, and multiple lines per species show selected thresholds from across the range of thresholds tested. Species are indicated by line colour, and are here shown for the environmental context with high variability/uncertainty and moderate starting-population sizes (SID 2). Results for other scenarios and including variability are in S2.1 Section in S2 Appendix.

between evaluation contexts. For proportional harvest strategies, differences in optimal harvest proportions were most definitively linked to species life history, with higher proportion optimal for faster species. While larger initial population sizes tended to allow larger proportions, this was not always the case (Fig 5B). In contrast to the constant harvest strategy, there were also clear differences between the different evaluation contexts in term of optimal harvest rates. For the threshold-proportional harvest strategy, optimal harvest parameters (both thresholds and proportions) showed substantial sensitivity to all environmental and evaluation factor contrasts (Fig 5C and 5D). This likely reflects the flexibility of this strategy to be tailored to different (conflicting) stakeholder interests, in contrast with the constant harvest strategy

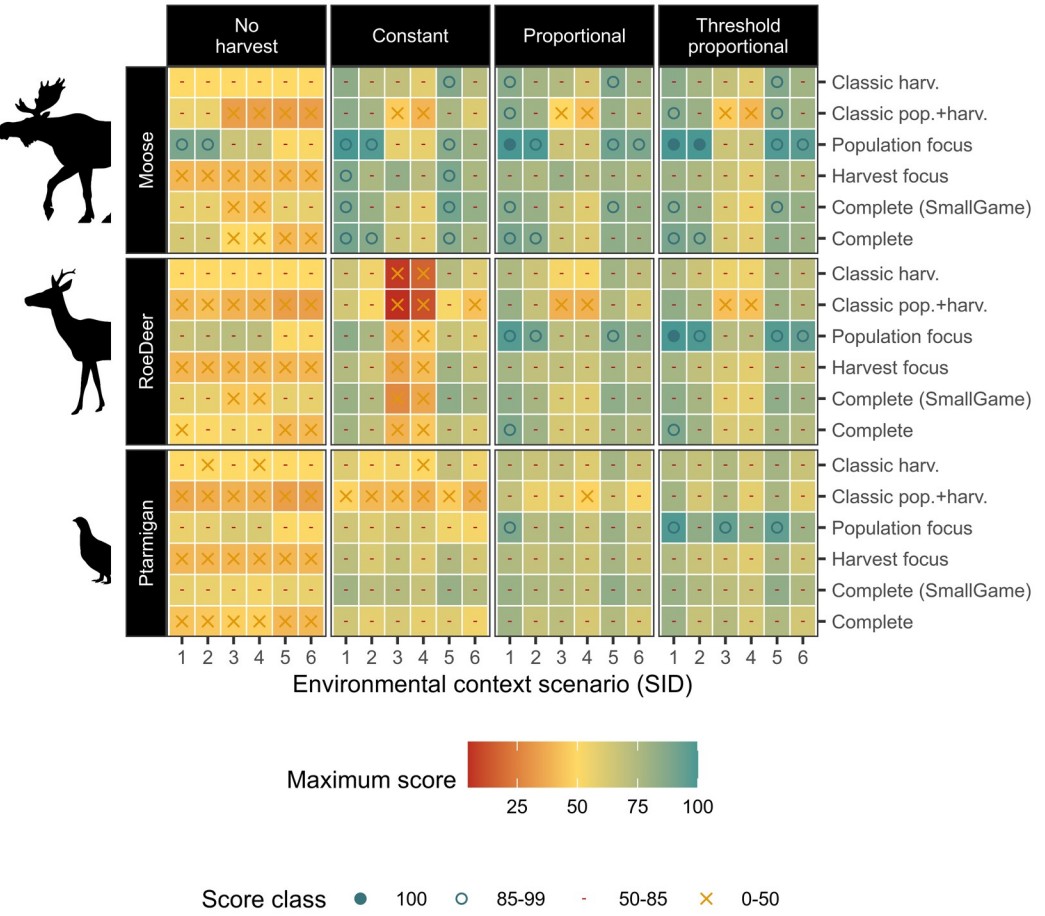

**Fig 3. Composite scores across harvest strategies and contexts.** Composite scores (colour) for each environmental (x-axis, and panel rows) and evaluation context (y-axis), under each harvest strategy (panel column), assuming harvest parameters are optimised under each harvest strategy. Environmental contexts (SID) codes are provided in Fig 1. Score classes (symbols) highlight where scores are maximal (i.e. 100, achieving the highest theoretically possible scores across all included metrics). This figure shows how scores are typically higher for slower life history species (e.g. *moose*) than faster life history species (e.g. *ptarmigan*), and for more complex harvest strategies (e.g. *threshold proportional*) than simpler strategies (e.g. *constant harvest*), but also that few maximal scores were found across the simulated contexts. Differences are further summarised in text and the following figures.

which has a relatively narrow sustainable operating range that is primarily environmentally determined, leaving low flexibility to cater for social preferences.

## Optimum harvest strategies

After optimizing the harvest parameters for each strategy and context, our simulations show that there was no universally optimum harvest strategy across all environmental and evaluation contexts (Fig 6). In fact, all harvest strategies could be perceived as an optimal choice in at least one environmental and evaluation context (Fig 6). However, in the evaluation contexts *Population focus*, *Classic pop.+harv.*, and *Classic harv.* a constant harvest strategy is never identified as optimal. In contrast, for *Harvest focus*, *Complete (small game)* and *Complete* composite set contexts, constant harvests are identified as optimal in 10 of the respective 18 environmental × evaluation cases for moose, as well as 2 cases in roe deer and once for ptarmigan (Figs 6 and 7).

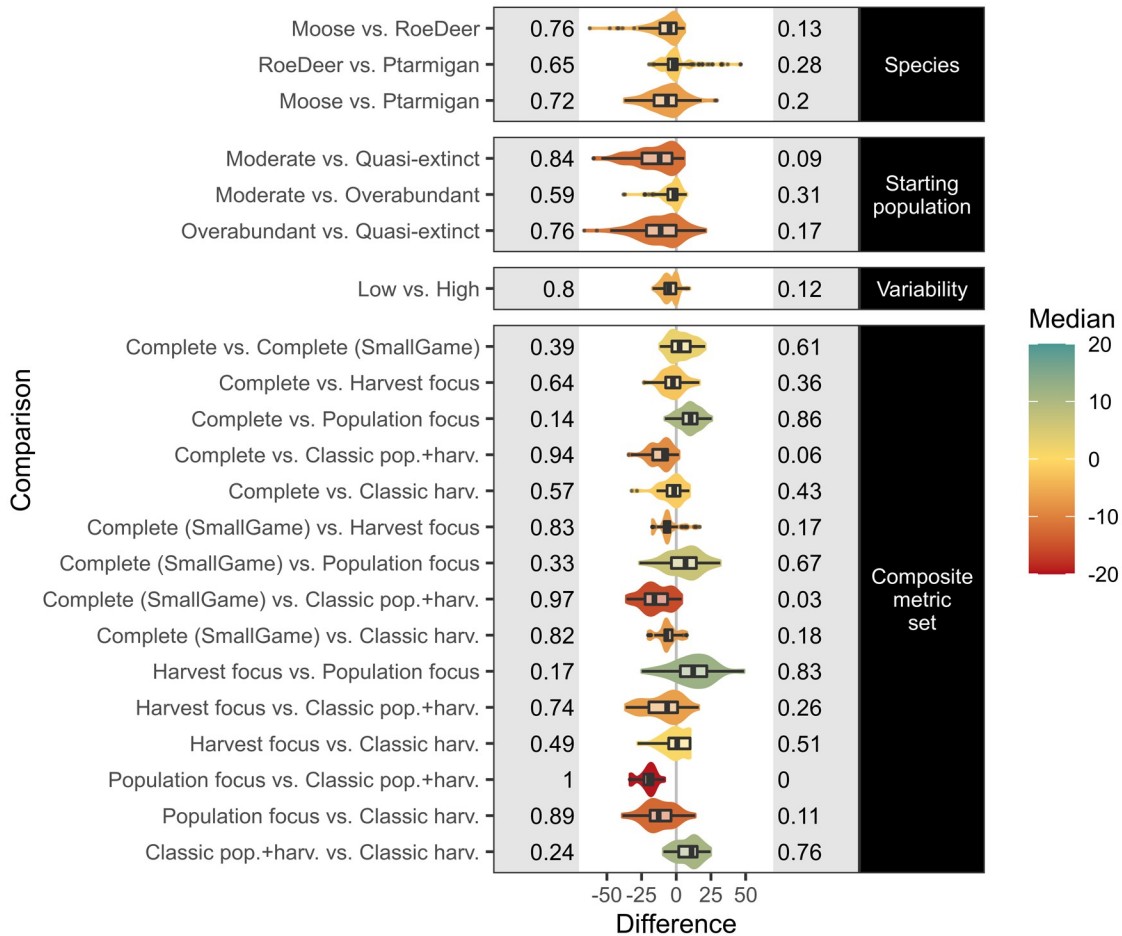

**Fig 4. Influence of environmental and evaluation factors on composite scores.** Differences in composite score outcomes (x-axis) due to differences in environmental and evaluation factors (y-axis), with all other factors held at equivalent levels for each pairwise contrast. Contrasts are given as change in the left-hand level vs. right-hand level, for example, *moose* typically result in a higher composite metric score than *roe deer*, all other factors equivalent. Violins show the data distributions (i.e. the values of pairwise contrasts, across all environmental and evaluation factors such as composite metric sets, that are not the focus of the pairwise contrast), with the colour indicating the median. Boxplots show the median, the first and third quartiles, and the whiskers extend to the smallest or largest value no further than 1.5 times the inter-quartile range from the hinge, with outliers plotted as points. Proportions of the observations below or above zero difference are given on the left and right grey panels respectively (and may not sum to one if some cases do not differ).

Overall, the most optimal harvest strategy was *threshold proportional*, which was optimal in 55.6% of the 108 species × environment × evaluation cases (and intermediate otherwise; Fig 6, comparing across panel columns). *Proportional* strategies were most often intermediate (57.4% of cases, with the remainder as best). In contrast, *constant* harvest strategies were optimal in only 12% of cases, and worst in 14.8%, while *no harvest* was an optimal choice in only 2 cases, and the poorest choice in 85.2% of cases.

Pairwise contrasts in environmental factors show that more complex harvest strategies are generally more preferable with faster life history species and higher variability scenarios (S2.4 Fig in S2 Appendix). For the more extreme starting populations, there were preferences towards both simpler and more complex harvest strategies, although most did not change. Pairwise contrasts between evaluation contexts show more definitive trends for many comparisons (S2.4 Fig in S2 Appendix).

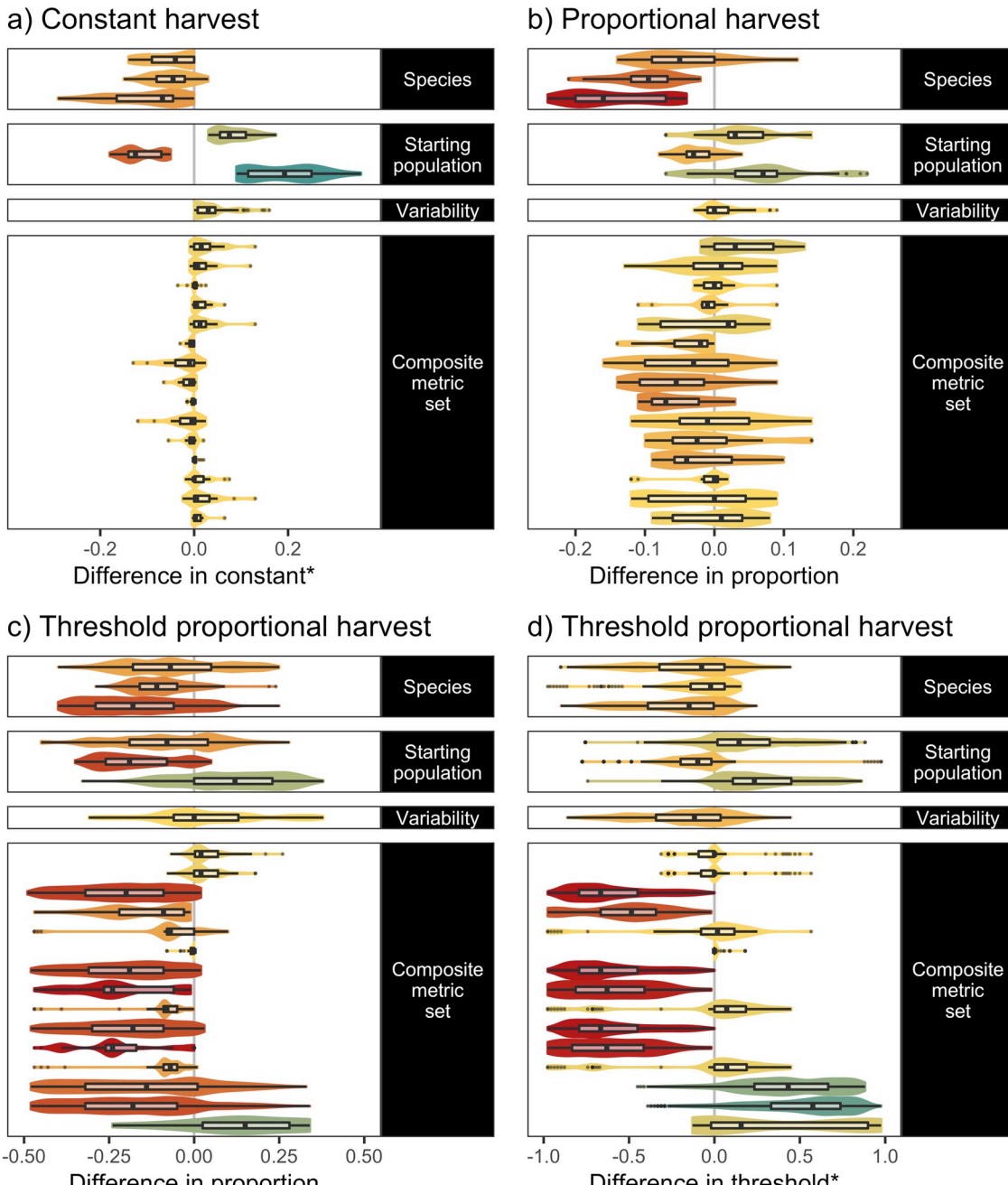

**Fig 5. Influence of environmental and evaluation factors on optimal harvest parameters.** Thumbnail figure (full figures given in the S2.3 Section in S2 Appendix) showing pairwise differences in optimal harvest parameters (i.e. those parameters giving the highest composite metric scores in each respective context) given environmental and evaluation factor contrasts, for a) constant, b) proportional, and c) and d) threshold proportional harvest strategies (proportion in c) and threshold in d) respectively). For further plot description, including colour legend, see Fig 4. The * indicates that the constant and threshold are scaled by the number of individuals considered as a 'moderate' population size for each of the species (i.e *Moose* = 600, *Roe Deer* = 6950, *Ptarmigan* = 17500). This overview figure shows how harvest parameter decisions are influenced relatively more by evaluation contexts in more complex harvest strategies (e.g. threshold proportional), compared with a simple 'constant harvest' strategy.

### Improvement in decision-making through use of environmental and evaluation context-specific heuristics

Without consideration of the environmental or evaluation contexts, the best choice for harvest strategy was *threshold proportional*. This would be the correct optimal choice in 55.6% of the 108 species × environment × evaluation cases, and result in an expected value forgone of 1.19% (Figs 6 and 7). *Proportional*, *constant*, and *no harvest* strategies would result in a mean value forgone of 2.75%, 12.2%, and 27.0% respectively.

Information on environmental contexts resulted in few improvements over the baseline of no contextual information. Use of species information resulted in an optimal decision in 59.3% of cases (with expected value forgone of 0.92%), selecting proportional for *moose* (optimal in 47.2% of cases, with expected value forgone of 1.64%), and *threshold proportional* for *roe deer* and *ptarmigan* (optimal in 61.1% and 69.4% of cases, with and expected value forgone

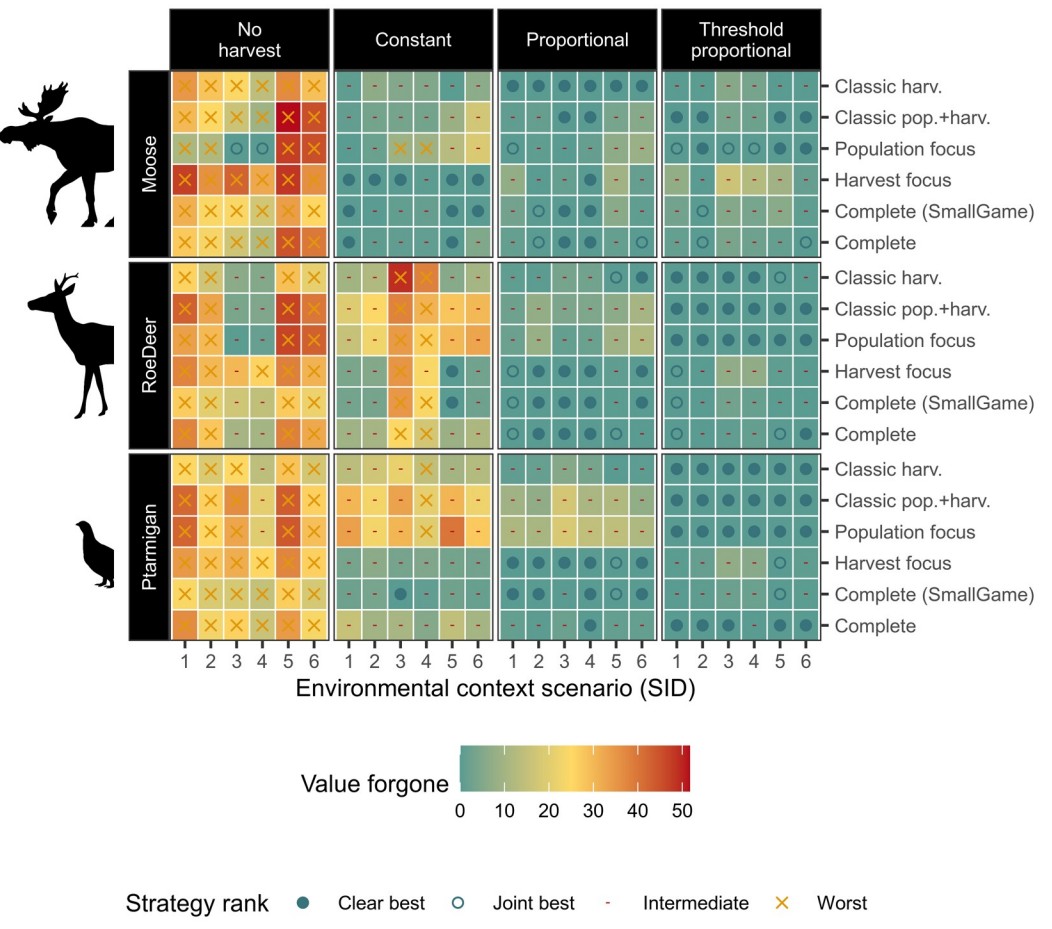

**Fig 6. Optimal strategy, and value forgone through choice of harvest strategy, across environmental and evaluation contexts.** Harvest strategy optimality (symbol), and value forgone (tile colour) by using the harvest strategy in each environmental and evaluation context, instead of the optimal strategy for the respective environmental and evaluation context. Harvest strategies (panel columns) are represented by their optimal harvest parameter outcomes. Environmental contexts are combinations of species type (panel rows), and starting population and variability (SID codes are described in Fig 1; x-axis). Strategies are compared for optimality only within equivalent species x environmental x evaluation comparisons. This figure shows how proportional and threshold proportional strategies are typically the most optimal, and typically result in lower value forgone when not. Differences are further summarised in text and the following figures.

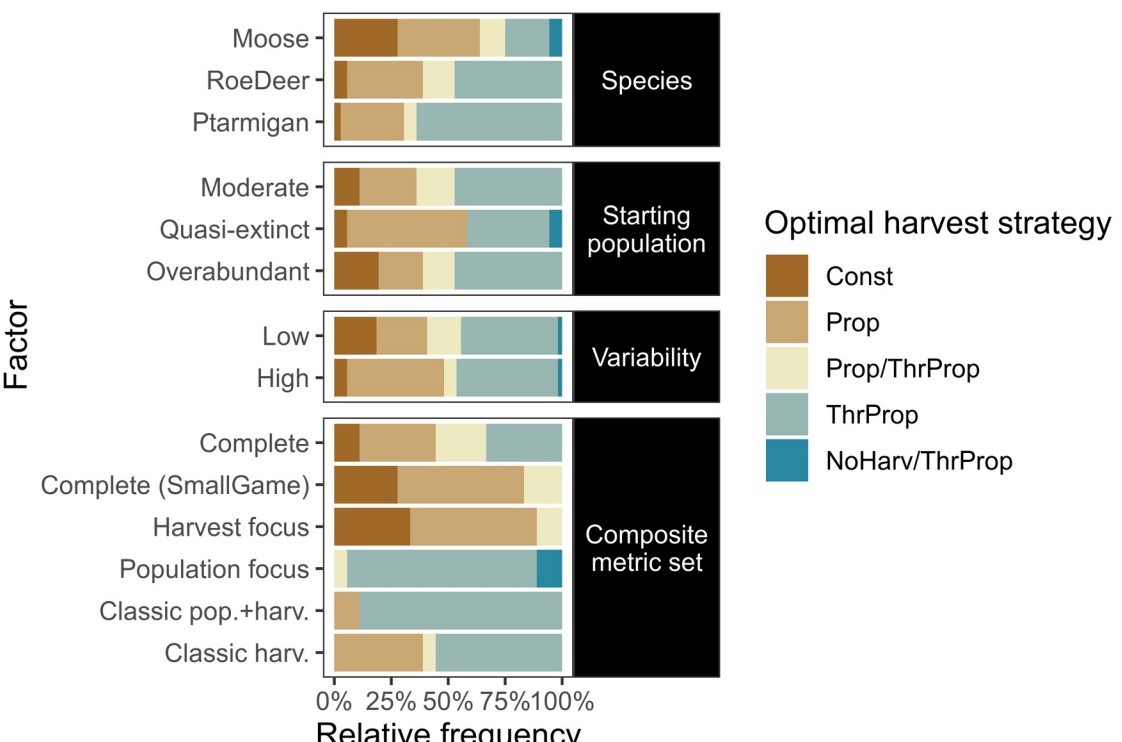

**Fig 7. Optimal strategy across environmental and evaluation contexts.** Relative frequency of optimal harvest strategy (or jointly optimal strategies) by environmental and evaluation context factors (y-axis). Each bar summarises the simulations including the factor specified on the y-axis. Optimal strategies are determined by ranking their respective best performing harvest parameter levels across harvest strategies. Pairwise comparisons between contexts are given in S2.4 Section in S2 Appendix.

of 0.58 and 0.54% respectively). Starting population information also improved decisions in 3.7% of cases compared to no information (expected value forgone 1.17%), and suggested *proportional* when population sizes are initially very low (at quasi-extinction; optimal in 52.8% of these cases, expected value forgone of 2.58%), and *threshold proportional* otherwise (optimal in 63.9% of cases with *moderate* starting population sizes, and 61.1% of cases with *overabundant* starting population sizes, with expected value forgone of 0.39 and 0.56% respectively). Information on variability level did not result in a change in strategy choice. *Threshold proportional* was optimal in 59.3% of *low* variability cases, and 51.9% of *high* variability cases, with expected value forgone of 1.53 and 0.86% respectively.

If all environmental context information was considered (i.e. decisions made based on species *and* environmental context, but not evaluation context), optimal decisions could be made in 63.9% of cases, with an expected value forgone of 0.57%. A *constant* strategy was selected for a third of the *moose* contexts (specifically, for *moderate* or *overabundant* starting populations, with *low* variability only), however this would be optimal in only half the cases within, and a *threshold proportional* strategy was preferable for the latter when aiming to minimise value forgone. *Proportional* was selected for *moose* contexts starting at *quasi-extinction* (optimal in 66.7% and 83.3% of cases for the *low* and *high* variability scenarios, respectively), and was selected as jointly optimal for 2/6 of the *roe deer* contexts, and one *ptarmigan* context (and therefore optimal in only half the cases within). *Threshold proportional* was optimal in all cases for *roe deer* with *moderate* starting populations and *low* variability, and for *ptarmigan* with *overabundant* starting populations and *low* variability, but for the remaining cases would

provide between 50–66.7% optimality. Decision-making based on minimising value forgone dropped *proportional* and *threshold proportional* from being jointly preferable in three and two environmental contexts, respectively.

In contrast, information on evaluation context could result in optimal decisions in 74.1% of cases (with an expected value forgone of 0.49%). This suggested a *threshold proportional* strategy for *Population focus*, *Classic pop.+harv.*, and *Classic harv.* composite metric sets (100%, 88.9%, and 61.1% of the respective cases). For *Complete (small game)* and *Harvest focus* composite metric sets, a proportional strategy is preferred, optimal in 72.2% and 66.7% of respective cases. For the *Complete* composite metric, either a *proportional* or *threshold proportional* strategy would be optimal in 55.6% of cases, but the *threshold proportional* strategy would result in a lower expected value forgone.

## Discussion

Aiming to develop heuristics for sustainability in wildlife harvest systems, we ran 289,848 stochastic models simulating harvest management under diverse environmental and evaluation contexts. The scarcity of contexts across our simulations resulting in high overall composite metric scores demonstrates the inherent complexity of achieving sustainability in terrestrial wildlife harvest systems with diverse stakeholders objectives [3, 4]. This large potential for conflicts and trade-offs emphasises that wildlife harvest decisions are likely to benefit from tools designed for decision-making under conflict and complexity. These tools include MSE models that can be used to evaluate and compare outcomes for multiple models, actions, and metrics [42, 43, 55], and Structured Decision Making (SDM) tools for management of conflicts through stakeholder negotiations [5, 56]. Avoiding exacerbating conflicts is endorsed in environmental management [57], and our analysis demonstrates how MSE can be used to map out conflict potential, and thereby contribute to conflict-sensitive stakeholder engagement.

Our results confirm that adaptive harvest systems such as proportional harvest, and particularly threshold-proportional harvests, were generally more likely to deliver good outcomes and be perceived as more sustainable. Adaptive harvest systems were higher scoring in more varied contexts, involved a less precipitous risk of population declines compared to constant harvest, and, result in the lowest levels of value forgone. This supports prior analytical and review comparisons showing general preference towards these adaptive strategies [15, 39, 58], including as a precautionary approach under uncertainty and variability, and importantly, extends systematic assessment across a diversity of environmental and evaluation contexts likely to be encountered in applied wildlife harvest management.

We found that no single harvest strategy was optimal across all environmental and evaluation contexts tested, however. Every harvest strategy was optimal in at least one case in every environmental context (Figs 6 and 7). The overall best strategy, threshold proportional harvest, was optimal in only 55.6% of cases evaluated. This supports the literature review findings of Derboa and Bence [39] in that optimal harvest strategies can differ depending on environmental contexts and evaluation procedures. In our simulations, information on environmental context (represented in this study as species, variability, and starting population size) could improve decision-making to be optimal in 63.9% of cases. Information on the evaluation context was more valuable, identifying optimal strategies in 74.1% of cases. There was large variation in outcomes of the harvest strategies when using different harvest parameters, however, and optimal parameters for suboptimal strategies can often score higher than suboptimal parameters for (potentially) optimal strategies (Fig 2). Information on environmental contexts was particularly influential in determining optimal harvest parameters in constant and proportional harvest strategies, while both environmental and evaluation context information were

influential for determining thresholds and proportions in a threshold-proportional harvest strategy (Fig 5). This likely reflects the superior ability of threshold-proportional strategies to be tailored to stakeholder perspectives, but simultaneously highlights the non-triviality of accounting for stakeholder perspectives in environmental management [9, 10].

The extent of the differences in outcomes across evaluation contexts suggests that, by focussing on limited evaluation metrics, prior theoretical analysis present a rather narrow and sometimes misleading perspective on the outcomes of harvest in socio-economically complex terrestrial wildlife systems. Differences due to composite metric sets were difficult to characterise, likely due to the interaction of the number and types of metrics included: more metrics can buffer each other and thus can increase scores, but can also increase the likelihood of trade-offs and thereby reduce mean scores. This emphasises the importance of identification and weighting of evaluation metrics in harvest management [56, 59, 60]. However, two key implications can be drawn from our results: 1) simpler 'classic' metrics commonly used in theoretical models may give a false perception of the magnitude of the benefits of more complex harvest strategies over constant harvests in some cases, and 2) the formulation of harvest objectives has a strong influence in determining optimal harvest strategies and parameters. This is particularly important to consider in the context of terrestrial wildlife harvest, where there is seemingly a widespread tendency for the objective of maximizing yields to be included, which persists even in cases where extensive stakeholder and manager engagement do not indicate maximum yields as a universally valued objective, and even while recognising the strong trade-off between population stability and harvest goals [61, 62]. In all of our simulated species the critical thresholds for a socio-ecologically desirable population size specified for management evaluation during expert elicitation were often well below the corresponding theoretical maximum sustainable yield levels (S1 Appendix). Inclusion of yield maximization is likely due to the classic tradition of yield being the sole focus of 'sustainability' in wildlife harvest outside a complementary and low bar objective of persistence (for example in early fisheries 'maximum sustainable yield' models), despite development of more diverse definitions [8]. Perhaps in fisheries contexts of the past this may have seemed appropriate, but in contemporary, predominantly recreational, terrestrial wildlife harvest there is no *a priori* reason to value maximizing mean harvests above or even equally to other objectives, especially given the diversity of human-wildlife conflicts associated with high density populations of some of the harvested species [4].

Faster life history species and higher variability contexts (due to stochasticity and uncertainty) were generally associated with reduced scores (Figs 3 and 4), and typically a greater preference towards more complex, adaptive harvest strategies. Much emphasis within the harvest literature has been on variability (stochasticity and uncertainty), typically revealing reduced sustainability with higher variability [13–16]. In these cases, thresholds can be used as a buffer from extinction [15, 17]. Our results are in line with these prior studies, but we also detected some noticeable exceptions. Many of the exceptions in our pairwise comparisons are due to threshold based evaluation criteria: for example when increased variability allows some replications to cross desirable threshold criteria (i.e. stochastic resonance [63]), without causing equivalent crossing of undesirable criteria thresholds. Other exceptions were likely due to closer alignment of 'ideal' population sizes (i.e. socially preferable levels) with populations sizes delivering maximum yields (as was the case for *roe deer* in our study), or due to a lack of perceivable difference in strategy outcomes under more extreme starting population sizes.

Management of slower life-history species was typically easier, and generally yielded relatively high scores even under simpler harvest strategies. However, the risk of precipitous declines via choosing suboptimal constant harvest parameters was greater, and the potential to recover from such low populations should be considered. In faster life history species recovering from extreme low populations, harvest strategy trades off speed, magnitude, and likelihood of recovery with harvest early in the time period, a trade-off likely to depend on the productivity of the population

[64]. In slower life history species recovery from low population sizes could be lengthy, with very low possibility of harvest [65]. Overall, this supports adaptive harvest strategies (including proportional and/or thresholds) as a precautionary strategy, providing economic and ecological resilience of harvest under both scientific and environmental uncertainty, and particularly uncertainty in the face of directional threats such as climate change [65].

Given our aim of developing heuristics across a range of species contexts for a set of harvest strategies, we developed our model using a consistent but relatively simple population dynamics framework. We specified our MSE models as one closed-population harvested species, undifferentiated by age, sex, or spatially, with logistic growth and simple characterisations of uncertainty and variability. We included prey species only, and do not explicitly consider biotic interactions such as their natural predators (these are assumed as captured within the variable demographic rates). We applied single decision rules over the whole time frame, and had no time-discounting or monetary valuation of costs and benefits, and a simplistic translation of outcomes into stakeholder values and utilities. We discuss these issues as they pertain to this analysis more in the full model description in the S1 Appendix. Further developments of the model would be valuable to extend the results to more complex and realistic contexts, in particular to examine impacts of different distributions and particular sources of uncertainty and variability (e.g. of population structuring during low abundances and infrequent extreme events [40, 66–68] and directional biases (e.g. in monitoring, quota setting, and harvest [19, 69]). We also do not consider starting conditions for stakeholders, such as current entitlement to harvest, which serves to frame outcomes as losses or gains. Current entitlement levels can severely constrain management decisions in practice [5], for example if Pareto improvements (no loss for any stakeholder) are emphasised in decision-making. While alternative assumptions may change the particulars of results, even the simple assumptions we employed resulted in many complex trade-offs among the diverse metrics evaluated, and we would expect the main conclusion of context dependency and importance of evaluation perspective to hold.

## Conclusions

Sustainability is a central, but often elusive goal of wildlife harvest management, challenged by complex socio-ecological systems, with many potential conflicts and uncertainties. Our stochastic simulation analysis provides the first detailed and consistent comparison of multiple sustainability metrics, across a representative range of common terrestrial wildlife game harvest systems. While we conclude, similarly to prior studies, that adaptive harvest systems including thresholds and proportional harvest were more likely to be perceived as sustainable in more variable contexts compared to constant harvest, our analysis reveals the many exceptions to this heuristic. Indeed, every harvest strategy was found to be optimal in every environmental context under at least one evaluation context. We found that the strongest driver of outcomes, optimal harvest parameters, and strategies was the evaluation context (i.e. the set of metrics used), rather than environmental contexts. However, adaptive strategies led to the least potential value forgone, and are likely a better risk-adverse strategy to employ to avoid low population sizes, which are likely to give poor outcomes for all stakeholders. Key implications for applied management are, first, that outcomes based on simplified metrics (e.g. persistence and maximizing mean harvest only) popular in the theoretical literature may give misleading impressions of the relative benefits of different harvest systems in complex socio-ecological systems. Second, while a threshold proportional strategy remains the optimal strategy across the majority of cases, both environmental and evaluation contexts have substantial influences on the optimal harvest parameters within this strategy. Our results highlight that trade-offs between sustainability objectives are largely inevitable, and, with no single optimum

strategy, 'optimal' harvest systems need to be identified with careful consideration of the appropriateness of sustainability metrics. Overall, heuristics derived from semi-complex MSE models such as this provide a useful bridge between over-simplistic theoretical models and complex context-specific models. We showed the potential of such heuristics to improve applied decision-making in low information contexts, and they are also likely to prove useful for guiding context-dependent sensitivity analyses in high information contexts, and the appropriateness of cross-context empirical comparisons.

## Supporting information

**S1 Appendix. Additional methods.**
(DOCX)

**S2 Appendix. Additional results.**
(DOCX)

## Acknowledgments

We thank Erling Solberg for discussions on moose management.

## Author Contributions

**Conceptualization:** Elizabeth A. Law, John D. C. Linnell, Bram van Moorter, Erlend B. Nilsen.

**Data curation:** Elizabeth A. Law.

**Formal analysis:** Elizabeth A. Law.

**Funding acquisition:** John D. C. Linnell, Bram van Moorter, Erlend B. Nilsen.

**Investigation:** Elizabeth A. Law, Erlend B. Nilsen.

**Methodology:** Elizabeth A. Law, Erlend B. Nilsen.

**Project administration:** Elizabeth A. Law.

**Software:** Elizabeth A. Law.

**Supervision:** John D. C. Linnell, Bram van Moorter, Erlend B. Nilsen.

**Validation:** Elizabeth A. Law.

**Visualization:** Elizabeth A. Law.

**Writing – original draft:** Elizabeth A. Law.

**Writing – review & editing:** Elizabeth A. Law, John D. C. Linnell, Bram van Moorter, Erlend B. Nilsen.

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
