## [Decision Letter · Decision Letter 0]

5 Aug 2021

PONE-D-21-14753

Sustainability of wildlife harvest in stochastic social-ecological systems

PLOS ONE

Dear Dr. Law,

Thank you for submitting your manuscript to PLOS ONE. After careful consideration, we feel that it has merit but does not fully meet PLOS ONE’s publication criteria as it currently stands. Therefore, we invite you to submit a revised version of the manuscript that addresses the points raised during the review process.

Please consider the comments from both Reviewers. In particular, please address the suggestion to define sustainability from Reviewer 1. Given that this term is ill-defined (line 48), how do we know that this paper captures it, as suggested by the title? Also, please discuss alternative ways to combine information across performance metrics (e.g., MinMax, or minimax), as suggested by Reviewer 2.

Also, please consider the following comments and editorial suggestions.

Much of the focus of MSE in fisheries has moved away from identifying single best harvest strategies to identifying those that “satisfice” stakeholder criteria to avoid over-reliance on models assumptions that often do not reflect reality (e.g., Harford and Carruthers 2017,). This paper diverges from that approach because of its focus on identifying heuristics for data-limited contexts. I suggest mentioning how and why this paper diverges from the current paradigm in fisheries MSE in the Introduction.Did you consider ecological objectives. Are these species important prey or predators? I suggest describing how the performance metrics (evaluation metrics) chosen are a subset of a much larger set of possible metrics, including those related to income/jobs from harvest which are not directly related to number harvested and depend on the management strategy.Why is performance poor on all metrics with no harvest? (even population persistence scores)? (Fig. 3)Why is performance poor for moderate, but not high and low environmental variability for Moose (and some extent Roe Deer)? (Fig. 3)Results depend on the caveats and assumptions of the model as partly explained in S1. Given that these assumptions are largely untested, they should be brought forward to the Discussion. For example, how realistic is the performance metric on extinction given limited knowledge about population dynamics at low abundances (i.e., allele effects).  Also, alternative distributions of population variability could be considered with longer tails to account for infrequent extreme events that strongly impact on performance (e.g., Anderson et al 2017). This can be included as a caveat in S1.1, and brought forward to the main discussion as well. In addition, biases in observation errors and harvest implementation can significantly impact performance, yet are not included and can be highlighted in the Discussion.Given the sensitivity of results to environmental variability, evaluation context, etc. and all the caveats of missing components/assumptions highlighted in S1 and by Reviewer 1, the conclusions and abstract should highlight the need for adaptive *and precautionary* management. This is especially true under climate change where future scenarios may not resemble historical conditions/trajectories or those simulated in the MSE.

Editorial suggestions

The difference between single composite metrics and holistic metrics is not immediately clear in the abstract on first read. I suggest giving brief examples in the abstract to illuminate your point.Is 1000 MC trials enough for risk-based performance metrics, especially those that pertain to low frequency events like extinction?How were the individual sustainability metrics chosen? (based on previous experience with management of those 3 species in particular, review of the literature, etc.?)The “Complete” composite score weights all metrics equally, so is a function of the number of metrics on each category. It would be more transparent to label it as equal weighting to population and harvest. “Complete” may be mean different things to different stakeholders.I suggest adding a description of the range of harvest parameters to the main text. It’s not clear how 289, 848 model runs are generated (line 213) from the text alone.In Figure 4, top three panels, is the composite score comparison for the complete set?Line 259, “result in lower composite metric scores, indicating stronger conflicts between objectives”. Can lower composite metric scores occur simply because of low score on all component metrics, i.e., without conflict?Please refer to Table S1.3 in caption or lead up to Fig. S2.1.1 to define SID1, SID2, etc…The 6 SID plots within Fig S2.1.1 look the exact same, despite representing different environmental contexts. This is not expected given the sensitivity of performance to environmental context in Fig. 3. The same for Fig. S2.1.2. The figures are not consistent with the text, “Optimum harvest parameters (that maximize the composite metric score) varied across environmental and evaluation contexts (Supplementary S2).” (lines 280-1).Fig S.2.2.1 and S2.2.3 Caption/text need to be added. Currently state “Description to be added.. These show the parameter levels that result in top scores (black outline) or within 5% of this top score (coloured areas).” (S2.2.1) and “Description to be added.. These show the parameter levels that result in top scores (black outline) or within 5% of this top score (coloured areas).” (S2.2.3).Suggest renumbering Fig S.2.2.3 as S2.2.2 is missingLine 331 “Pairwise contrasts in environmental factors show that more complex harvest strategies generally become more preferable with faster life history species and higher variability scenarios (Supplementary Fig S2.4).”  This is not obvious from Fig. S2.4. Moving from Moose to Ptarmigan, it looks like there is a high frequency of shift to simpler optimal  harvest rules. Moving from low to high variability it looks like optimal harvest rules generally stay the same in complexity. Figure S2.4 needs a caption and description.Line 339. Can you clarify what “cases” means in this sentence (all scenarios within 6 harvest rules, optimal harvest parameters, 6 SIDs, 3 species?), “This would be the correct optimal choice in 55.6% of cases, and result in an expected value forgone of 1.19% (Fig 6-7). Proportional, constant, and no harvest strategies would result in a mean value forgone of 2.75%, 12.2%, and 27.0% respectively”Line 354, Please clarify that these results aggregate over evaluation contexts and species with optimal harvest parameters “If all environmental context information was considered, optimal decisions could be made in 63.9% of cases, with an expected value forgone of 0.57%.”

We look forward to receiving your revised manuscript.

Kind regards,

Carrie A. Holt

Academic Editor

PLOS ONE

2. Thank you for stating the following in the Acknowledgments Section of your manuscript.  (Auhtor’s contribution)

“Conceptualisation and writing-review (EL, JL, BvM, EN), data curation, formal analysis, validation, visualisation and writing-draft (EL), investigation and methodology (EL, EN), funding acquisition and supervision (JL, BvM, EN).”

“This study was funded by the Research Council of Norway (https://www.forskningsradet.no/; grant 251112; JL, BM, EN). The funders had no role in study design, data collection and analysis, decision to publish, or preparation of the manuscript.”

Please include your amended statements within your cover letter; we will change the online submission form on your behalf."

Additional Editor Comments (if provided):

Reviewers' comments:

Reviewer's Responses to Questions

**Comments to the Author**

1. Is the manuscript technically sound, and do the data support the conclusions?

Reviewer #1: Partly

Reviewer #2: Yes

2. Has the statistical analysis been performed appropriately and rigorously? 

Reviewer #1: Yes

Reviewer #2: Yes

3. Have the authors made all data underlying the findings in their manuscript fully available?

Reviewer #1: Yes

Reviewer #2: Yes

4. Is the manuscript presented in an intelligible fashion and written in standard English?

Reviewer #1: Yes

Reviewer #2: Yes

5. Review Comments to the Author

Reviewer #1: I have reviewed the manuscript entitled “sustainability of wildlife harvest in stochastic social-ecological systems” prepared by Law et al. The authors use a MSE framework to simulate the effects of harvest on wildlife under different combinations of environmental, harvest and evaluation contexts. In general, the paper is well written and presented with a lot of simulation work supporting the results. However, I do have a few concerns that need to be addressed before considering for publication in PLOS ONE.

1. The title needs to be more informative. The current title is fairly broad and vague and it’s difficult for the readers to capture what the authors are trying to address in this paper. Furthermore, I don’t think the simulation work done in this paper is adequate to address this broad topic.

2. The purpose of this paper is not quite clear to me. The authors used MSE to simulate and compare outcomes under different combinations of environmental, harvest and evaluation contexts, and then basically conclude the outcome is context-dependent (isn’t this something self-evident?). There are numerous combinations in the simulation and I don’t quite get what is the focus/main question. It is important to have a hypothesis-driven topic in scientific papers, and I encourage the authors to further think about this and reorganize the paper on some specific question/hypothesis/prediction.

3. For the environmental context (others may refer to operating model in MSE), the authors used logistic model for the population dynamics with 18 combinations of 3 species, 2 levels of variability and 3 starting population sizes. The variability levels apply to 4 parameters (r, m, q and h), so there should be 2^4=16 scenarios of variability, unless the authors assume the 4 parameters are inter-correlated (if so, why?).

4. To introduce stochasticity in the simulation, the authors assume normal distribution of the 4 variable parameters, which is likely to generate negative values. To avoid negative values of abundance et al., a method “round to nearest positive value” is used, but this will generate bias in the simulation. A better practice is to use log-normal distribution to introduce stochasticity.

5. For the evaluation context, the authors need to further explain how these individual and composite metrics are related to definitions of “sustainability”. If sustainability is the theme of this paper, the authors should try to give their own definition upfront, so that they can better quantify this term in the evaluation context.

6. When calculating composite metrics, do you give equal weight to individual metrics? If so, how do you balance the number of individual metrics related to population or harvest?

7. In the result section, different composite metric score reflects different perspective on sustainability, and I’m not sure how this can be summarized across composite metrics. In line 234, what is the meaning of 85% of composite metric score? Does this reflect different things among composite metric scores. Line 236, “better performing context” is a weird statement, so some environment/evaluation contexts are better than others?

8. The resolution of figure 1 is a bit low.

Reviewer #2: I had the chance to review the paper “Sustainability of wildlife harvest in stochastic social-ecological systems” for consideration as a Research Article in PLOS ONE. Overall, I really like this paper and recommend that it is suitable for publication with some minor revisions. Great job! The focus of this paper is on using a robust combination social, management, and ecological scenarios of a harvested ecosystem to provide general rules of thumb for decision-making in complex systems. I found the model competently executed, the scenarios explored were interesting, and the paper is very well written. I don’t have many major comments or critiques to the paper. The main comments that I have are on interpreting the visuals and the Discussion text and begin in no particular order below:

Did the authors consider some sort of minimax (MinMax) decision rule to evaluate which harvest strategies minimized the maximum possible loss among all scenarios considered? In this case, the loss function could be the composite score or the average value forgone. I think this would provide a very generalizable heuristic on the decision rules that help to achieve an acceptable level of value across the uncertain range of management and ecological contexts. Is this already captured in Figure 7 with the ‘optimal strategy’?

What is meant by L. 75 deriving general inferences from case studies? Do you mean that we lack heuristics from empirical case studies because these case studies aren’t experimental? What about meta-analyses?

Table 2: in the composite metric set, are the individual metrics weighted equally or unequally? I think L. 200 suggests they are equal. In some fisheries literature (Carruthers et al. 2018; van Poorten & Camp 2019) there is consideration that some metrics should be weighed quite differently (e.g., Persistence weighed 2x more than Harvest Mean). Or, in other words, management may only really care about Persistence when it is low (near some conservation-relevant levels) but not when its high, otherwise we care more about the Harvest Mean (or other criteria).

Carruthers, T. R., Dabrowska, K., Haider, W., Parkinson, E. A., Varkey, D. A., Ward, H., ... & Post, J. R. (2019). Landscape-scale social and ecological outcomes of dynamic angler and fish behaviours: processes, data, and patterns. Canadian Journal of Fisheries and Aquatic Sciences, 76(6), 970-988.

van Poorten, B. T., & Camp, E. V. (2019). Addressing challenges common to modern recreational fisheries with a buffet-style landscape management approach. Reviews in Fisheries Science & Aquaculture, 27(4), 393-416.

I really like the description of the model methods and procedure. Nicely done.

I found several of the Figures difficult to make intuitive sense of, particularly Figures 3, 4, 5, and 6. As these are the main results Figures, I suggest either some changes to the figures, the figure captions, or the accompanying text in the main Results to help the reader make these more intuitive to take inferences from. Figures 3 and Figure 6 are quite similar but showcase different metrics (composite scores versus value forgone) so my comments pertain to both.

For Figure 3/6: I am unsure on how the ‘rank’ symbols are meant to be inferred and how the reader is supposed to compare and contrast rows vs. columns vs. colours vs. symbols. Let’s take Figure 6 for example: for Moose managed under Threshold Proportional with a Population Focus: there are 3 ‘clear bests’ outcomes across the environmental contexts and within an environmental context (for example Context 2) there are 2 clear bests and 2 joint bests strategies. What is the main takeaways from this figure that you are trying to communicate to the reader? I am not sure if it is clear how the reader is supposed to read and compare the scenarios. It looks really great though!

For Figure 4: I think it shows the difference between composite scores between two scenarios (all else equal). But does a negative difference mean that the left-hand side had the higher score or the lower score? Example of some of my confusion can be illustrated in the first comparison Moose v. Roe Deer: across all scenarios, all else equal, a Moose life history had a negative difference in the composite score than a Roe Deer life history – so does that mean that Moose tended to have the higher score or the lower score? What does the 0.76 and 0.13 mean? And can you walk through what the takeaway here is for Moose, as an example? If I read this right and Moose have a negative difference in the composite score, is the Moose life history harder to manage (under the assumptions of this model) than a Roe Deer life history? I think based on Figure 3 that Moose are slightly easier to manage (likely because they are less variable), so I think part of my challenge can be helped by having the Figure caption or results text better walk through these figures.

For Figure 5: the figure caption discusses optimum harvest parameters. But is this meant to be a sensitivity test? Or what is it? How can you have a difference in the threshold/harvest constant when I think you have set the threshold as an input parameter? And what do the colors represent (there’s no color legend provided). When I see this figure and the axis labels, I though it was showing a sensitivity test but the boxplot seems to indicate it is the outcome of many simulations.

Some of the Discussion paragraphs read more like Results. For example, L. 392 mostly focuses on the paper results. I suggest broadening the Discussion out a bit more to engage with the ongoing conversations about adaptive management, monitoring, and decision-making from wildlife and fisheries papers.

6. PLOS authors have the option to publish the peer review history of their article (what does this mean?). If published, this will include your full peer review and any attached files.

Reviewer #1: No

Reviewer #2: No

---

## [Author Response · Author response to Decision Letter 0]

22 Sep 2021

Please refer to the "Response to Reviewers" file.

---

## [Editor Report · Decision Letter 1]

1 Oct 2021

PONE-D-21-14753R1Heuristics for the sustainable harvest of wildlife in stochastic social-ecological systemsPLOS ONE

Dear Dr. Law,

Thank you for submitting your manuscript to PLOS ONE. After careful consideration, we feel that it has merit but does not fully meet PLOS ONE’s publication criteria as it currently stands. Therefore, we invite you to submit a revised version of the manuscript that addresses the points raised during the review process.

Please consider the following editorial comments

**Line 191-2.** I suggest adding some additional text on these lines “While we aimed to include as broad a range of sustainability metrics as possible, these may not be adequately representative *of ecological or socio-economic objectives* that are not rooted in volumes or variability of either harvest or population sizes. *For example, objectives related to ecosystem function or economic equity may not be captured here*” (or something similar).

**Fig 3 **Would you consider putting the SID scenarios in order from low to high. I was confused in my earlier comments, as I assumed SID 3-4 represented moderate initial population sizes, being in the middle (not low).  The current order is not intuitive.

**Fig 4** caption. “Differences in composite score outcomes (x-axis) due to differences in environmental and evaluation factors (y-axis), with all other factors held at equivalent levels for each pairwise contrast”. Apologies, but this caption is still not clear to me. I suggest adding “across all composite metric sets” to clarify (as you mentioned in your reply to E12).

**Fig S2.1.1 and S2.1.2** Some text in the Supp. Mat. explaining Figures for SID 3 and 4 might helpful, e.g., describing how lines are often not visible because…

**Fig S2.4.** I suggest adding an explanation of the legend. In particular, “+ Simpler”, “-More Complex”, “- Simper”, and “+More Complex” are not intuitive.

**Discussion or S1 (Section on Caveats)**. I suggest adding future research that includes other distributions of random variables. I agree with Reviewer 1, that the normal distribution is not the most biologically appropriate approach, but arguably can be used here, at least provisionally, for this study on heuristics.

We look forward to receiving your revised manuscript.

Kind regards,

Carrie A. Holt

Academic Editor

PLOS ONE
---

## [Author Response · Author response to Decision Letter 1]

2 Nov 2021

Please see the response to reviewers document.

---

## [Editor Report · Decision Letter 2]

4 Nov 2021

Heuristics for the sustainable harvest of wildlife in stochastic social-ecological systems

PONE-D-21-14753R2

Dear Dr. Law,

We’re pleased to inform you that your manuscript has been judged scientifically suitable for publication and will be formally accepted for publication once it meets all outstanding technical requirements.

Kind regards,

Carrie A. Holt

Academic Editor

PLOS ONE
---

## [Editor Report · Acceptance letter]

11 Nov 2021

PONE-D-21-14753R2 

Heuristics for the sustainable harvest of wildlife in stochastic social-ecological systems 

Dear Dr. Law:

I'm pleased to inform you that your manuscript has been deemed suitable for publication in PLOS ONE. Congratulations! Your manuscript is now with our production department. 

Kind regards, 

on behalf of

Dr. Carrie A. Holt 

Academic Editor

PLOS ONE